# Coconut Mesocarp Extracts to Control *Fusarium musae*, the Causal Agent of Banana Fruit and Crown Rot

Jesús Aidmir Yeikame Morelia-Jiménez [1], Beatriz Montaño-Leyva [2,*], Francisco Javier Blancas-Benitez [1], Luz del Carmen Romero-Islas [1], Porfirio Gutierrez-Martinez [1], Luis Guillermo Hernandez-Montiel [3], Pedro Ulises Bautista-Rosales [4] and Ramsés Ramón González-Estrada [1,*]

1   I.T. Tepic, Tecnológico Nacional de México, Av. Tecnológico #2595, Tepic 63175, Nayarit, Mexico; jeaimoreliaji@ittepic.edu.mx (J.A.Y.M.-J.); fblancas@ittepic.edu.mx (F.J.B.-B.); lromero@ittepic.edu.mx (L.d.C.R.-I.); pgutierrez@ittepic.edu.mx (P.G.-M.)
2   Departamento de Investigación y Posgrado en Alimentos, Universidad de Sonora, Rosales y Blvd. Luis Encinas, Hermosillo 83000, Sonora, Mexico
3   Centro de Investigaciones Biológicas del Noroeste, Av. Instituto Politécnico Nacional 195, Col. Playa Palo de Santa Rita Sur, La Paz 23096, Baja California Sur, Mexico; lhernandez@cibnor.mx
4   Unidad de Tecnología de Alimentos—Secretaría de Investigación y Posgrado, Universidad Autónoma de Nayarit, Ciudad de la Cultura SN, Tepic 63000, Nayarit, Mexico; ubautista@uan.edu.mx
*   Correspondence: beatriz.montano@unison.mx (B.M.-L.); ramgonzalez@ittepic.edu.mx (R.R.G.-E.)

**Abstract:** Crown rot, caused by *Fusarium* species, is the most devastating postharvest disease in bananas. Fungicides are traditionally applied as a postharvest treatment to control crown rot in bananas. However, there is a need to research environmentally friendly compounds as postharvest treatments instead of chemical fungicides. The phenolic compounds gallic acid, protocatechuic acid, and chlorogenic acid were identified in coconut mesocarp extract. Overall, the treatments were more efficient in crown-based than fruit-based culture mediums. The mycelial development was inhibited in a range from 20 to 26% (applying coconut mesocarp extract at 5%) compared to the control. Sporulation and spore germination were significantly inhibited, with a reduction of 88% in spore production and 91% in spore germination inhibition compared to the control. In in vivo tests, the aqueous extracts were effective by limiting the percentage of infected fruit, crown rot, and fruit severity. The use of coconut mesocarp extracts can be an effective and environmentally friendly alternative to the use of fungicides for controlling *Fusarium musae* on bananas.

**Keywords:** SEM; *Musa* spp.; crown rot; gallic acid; protocatechuic acid; chlorogenic acid



## 1. Introduction

Banana (*Musa* spp.) is the most commercialized tropical fruit in the world [1]. However, the quality of the fruit and its postharvest life can be affected by the presence of phytopathogens [2]. Crown rot infection can occur at harvest time, and the appearance of mycelia on the surface of the crown is a sign of the presence of phytopathogens, with the subsequent mycelial development into the peduncles and eventually the fruit causing softening and necrosis leading to the detachment of the banana in severe cases [3].

Chemical control by spraying fungicides is the principal strategy for disease management in *Musa* spp. [4]. However, the excessive use of chemical fungicides has had negative effects on the environment and human health; therefore, research in this field is currently directed toward the search for effective alternatives for the control of phytopathogens [5]. One of these is the use of plant extracts as an environmentally friendly alternative against pathogenic fungi [6,7]. In a recent study, *Colletotrichum musae* isolated from banana was inhibited (100%) in vitro by the addition of leaf extracts from medicinal plants such as *Nyctanthes arbourtis* and *Acasia nilotica* [8]. Jehani et al. [9] found that the use of neem plant (*Azadirachta indica* L.) extracts inhibited the in vitro mycelial growth (73%) of *C. capsici*, a

pathogen of yam (*Dioscorea alata* L.). Recently, coconut mesocarp extracts were efficient in controlling *Penicillium italicum* infection in Persian lime fruits by reducing the disease incidence and severity on artificially infected fruits by spraying the extracts in a preventive mode during storage [10]. Authors related the antifungal activity with the presence of chlorogenic and hydroxybenzoic acids and gallocatechin in the aqueous extracts. Furthermore, in another study, coconut mesocarp extracts were incorporated as an additive into a chitosan matrix, and the results showed good biocompatibility of the extracts with the polymeric matrix by enhancing the physicochemical and antifungal properties of films and coatings, protecting tomato fruits against *Geotrichum candidum* establishment [11]. This study determined the antifungal capacity of coconut mesocarp extracts against *Fusarium musae* and their impact on fungal control in bananas.

## 2. Materials and Methods

### 2.1. Raw Materials

Coconut mesocarp tissue was obtained in San Blas, Nayarit, Mexico (21°32′27.4″ N and 105°17′28.7″ W). Banana fruits were purchased from a commercial orchard located at 21°28′28.6″ N and 104°51′31.1″ W in Tepic, Nayarit, Mexico. Potato dextrose agar (PDA) was purchased from Sigma-Aldrich (St. Louis, MO, USA). Formic acid and methanol were purchased from Sigma-Aldrich.

### 2.2. Isolation, Purification, Pathogenicity Test, and Molecular Analysis of Fusarium spp.

Healthy mature banana fruits were used for the isolation of the pathogen; to allow fungal infection, the fruit was placed in a chamber with a relative humidity between $90 \pm 5\%$ at ambient temperature. After 9 days, tissues (crown and fruit) with disease development were selected and cut ($1 \times 1$ cm) considering 50% of the infected area and 50% of the non-infected area. The selected tissues were then treated with sodium hypochlorite solution (NaClO at 2%) for 2 min, rinsed with sterile distilled water (SDW), placed on PDA plates, and incubated at 25 °C [12]. The isolated fungus was purified, and monosporic isolation was performed. Koch's postulates were then evaluated to determine the pathogenicity of *Fusarium musae* [12]. First, a spore suspension was prepared according to the methodology of Iñiguez-Moreno et al. [13]. Then, disease- and damage-free bananas were washed, disinfected, and artificially infected with the spore suspension (10 μL, $1 \times 10^6$ conidia/mL) of the pathogen, and the wound was made in the crown and fruit tissue using a sterile needle [14]. Control fruits were treated with SDW. Finally, fungal reisolation was performed with the same protocol to isolate *Fusarium musae*. For molecular identification, *Fusarium* sp. was grown in potato dextrose broth (PDB). DNA was extracted according to the method of Raeder and Broda [15]. The fungus was identified using the ITS1-5.8S-ITS2 region of the rRNA gene cluster, and the primers used were ITS1 (5′-TCCGTAGGTGAACCCTGCGC-3′) and ITS4 (5′-TCCTCCGCTTATTGATATGC-3′). The reactions were amplified using a thermocycler (Bio-Rad Laboratories, Hercules, CA, USA). The amplicons were sequenced in both directions and compared with the sequences of type strains in GenBank DNA database using the basic local alignment search tool BLAST.2.6 [16].

### 2.3. Obtention of the Aqueous Extracts and Chromatographic Analysis

The aqueous extracts of coconut mesocarp were obtained using the method proposed by Cortés-Rivera et al. [17], by mixing 0.5 g of mesocarp with 25 mL of SDW, later shaken for 1 h, and then centrifuged, to be later filtered with acrodisks (MilliporeTM, 0.45 μm, Darmstadt, Germany), to obtain the solutions of coconut mesocarp extracts. The extract solutions were stored in amber flasks prior to the experiments.

The identification of the profile of phenolic compounds was carried out using the method reported by Hernández-Flores et al. [10]. First, 0.1 g of the sample was dissolved in 5 mL of ultrapure water and vortexed for 1 h. Then, the sample was centrifuged (15 min at 5000 rpm) and filtered (0.45 μm membrane). Identification of phenolic compounds was performed in HPLC-DAD equipment (Agilent 1200, Santa Clara, CA, USA) using a Zorbax

Eclipse Plus C18 reverse phase column (4.6 mm × 100 mm) with a particle size of 3.5 μm. The injection volume was 10.0 μL, and the column temperature was 30 °C. A step gradient elution with absolute methanol (solvent A) and 1% formic acid (solvent B) was used: 18% A (0 min); 30% A (4 min); 45% A (8 min); 55% A (10 min); 70% A (11 min); and 100% A (13 min). UV detection was carried out from 214 to 520 nm. The retention times of the standards were used for the identification of phenolic compounds.

## 2.4. In Vitro Antifungal Assay

Banana (*Musa × paradisiaca*) tissues (crown or fruit) were used to prepare the culture medium and broth. The culture medium was prepared by adding 28 g of healthy fruit tissues (crown or fruit) and dissolving them in 400 mL of SDW. Then, the agar (8 g) (MCD LAB^TM Tlalnepantla de Baz, Mexico) was added to the solution and stirred for 1 h (500 rpm); subsequently, the culture solutions were sterilized at 121 °C for 15 min [18]. Thereafter, the culture solutions were filtered through sterile gauze to retain fruit tissues (crown or fruit).

The culture broth was made by adding 28 g of healthy fruit tissues (crown or fruit) and dissolving them in 400 mL of SDW and then stirring for 1 h (500 rpm); then, the broth solution was sterilized at 121 °C for 15 min [18]. Thereafter, the broth solutions were filtered through sterile gauze to retain fruit tissues (crown or fruit).

Finally, the previously prepared coconut mesocarp extracts were added to the culture media at ambient temperature in a biosafety hood.

For the determination of mycelial growth inhibition, the plug method was used with some modifications [19]. PDA plugs (10 mm in diameter) from 6-day-old cultures of the pathogen were cut. Subsequently, plugs containing the tissue-based culture medium supplemented with the extracts (1, 5, 10% *v/v*) were cut and replaced with plugs containing the fungus. Inoculated plates were incubated at 28 °C for 6 days, and the colony development was measured daily using a digital Vernier (Truper^TM, Culiacán, Mexico). Control plates consisted of tissue-based culture medium (crown or fruit) without extract solutions. The results were expressed as a percentage of mycelial growth inhibition compared to the control, using the formula proposed by Mejdoub-Trabelsi et al. [20].

For the evaluation of germ tube elongation, 100 μL of the spore solution ($1 \times 10^6$ spores/mL) was inoculated into a tissue-based culture broth (1 mL) in Eppendorf tubes containing the extracts (1, 3, and 5% *v/v*). Control samples consisted of tissue-based culture broth (crown or fruit) without extract solutions. The samples were observed under the microscope (Motic, BA300, Hong Kong, China) after 12 h, 200 spores per sample were quantified, and finally, the germinated spores were calculated. For the sporulation test, Petri dishes where mycelial growth was evaluated were used. SDW (10 mL) was added to the fungal lawn and rubbed using a sterile inoculation loop, to release the spores from the fungal structures. The suspension was filtered through a sterile cheesecloth. A hemocytometer (Hausser Scientific®, Horsham, PA, USA) was used to calculate the spore concentration (number of spores/mL). Tests were performed in triplicate.

## 2.5. In Vivo Antifungal Assay

Prior to the inoculation and application of treatments, banana fruits were washed, disinfected (NaClO 2%, *v/v*) for 2 min, rinsed with SDW, and dried at room temperature (25 °C) [12]. Aqueous extracts were applied in a biosafety hood by spraying 1 mL on each fruit or SDW (control fruit) as appropriate. Subsequently, the treated fruit was dried (25 °C) for 1 h in a biosafety hood. The inoculation of the fungus was carried out as follows: each banana was wounded twice with a sterile needle (diameter 0.35 and 0.3 cm); except for the crown, only one wound was made (in the center of the crown). Thereafter, the wounds were inoculated using *Fusarium musae* with 10 μL of the spore suspension ($1 \times 10^6$ spores/mL). The fruits were placed in a storage chamber (Torrey®, San Nicolás de los Garza, Nuevo León) under the following conditions: 25 °C, 85–90% relative humidity for 6 days.

### 2.6. Analysis by Scanning Electron Microscopy

The microstructural analysis of the tissues (crown and fruit) treated with the aqueous extract was carried out after six days post-application of treatments. The protocol proposed by González-Estrada [21] was used for the preparation of the samples, and a scanning electron microscope MINI-SEM (SNE–3200M, South Korea) was used to obtain the micrographs.

### 2.7. Statistical Analysis

A bi-factorial design was used for the in vitro test. For mycelial growth and sporulation assays, five Petri dishes were used, and 5 Eppendorf tubes containing the tissue-based culture broth were used per replicate for germination. For in vivo tests, a uni-factorial design was used with 30 crowns and 30 fruits per replicate. In separate experiments, all tests were repeated twice. Analysis of variance (ANOVA) followed by Fisher's LSD test were used [21]. The level of statistical significance was $\alpha = 0.05$ using Minitab Statistical Software® (version 19.2020.1.0).

## 3. Results

### 3.1. Isolation, Pathogenicity Test, and Molecular Analysis of Fusarium sp.

The morphology of the *Fusarium* isolated from the banana is shown in Figure 1. A colony with a radial growth of six days old and a dense aerial mycelium predominantly violet pink in color with a white halo (Figure 1a) was observed; on the backside, the colony presented colors from violet to black violet with a yellow pale halo (Figure 1b). Single spores presented an ellipsoidal form (Figure 1c). In the pathogenicity test, fruit rot was evident after three days after inoculation by presenting a dark color in the inoculation zone; also, the fruit presented a softening area. At the end of the storage time (9 days), the crown rot and fruit presented an extensive necrotic area with the presence of a dense aerial white-gray mycelium (Figure 2a,b). The identification of the phytopathogenic fungi performed by PCR gave a size of 520 bp (Figure 1d). According to the comparison of the sequence of *Fusarium* sp. isolate with the GenBank database of the National Center for Biotechnology Information (NCBI, USA) [22], the isolated strain presented 100% identity with the fungus *Fusarium musae* with accession number OW986028 [23].

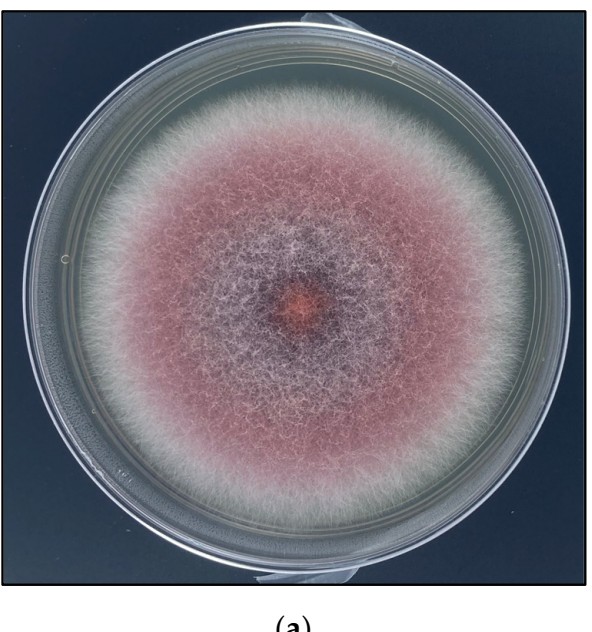

(**a**)

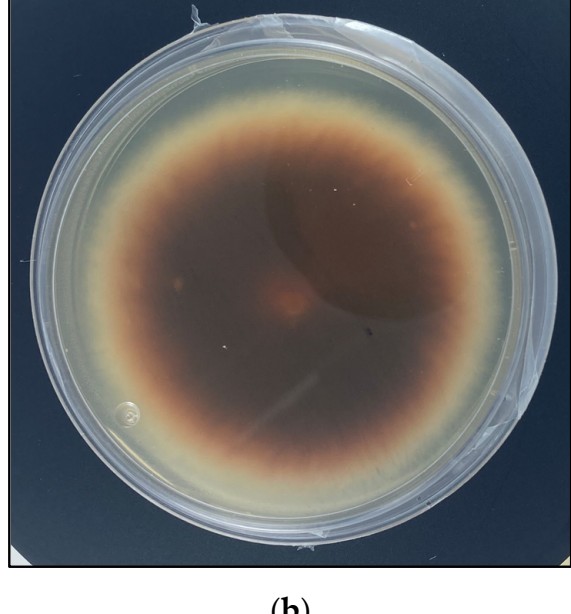

(**b**)

**Figure 1.** *Cont.*

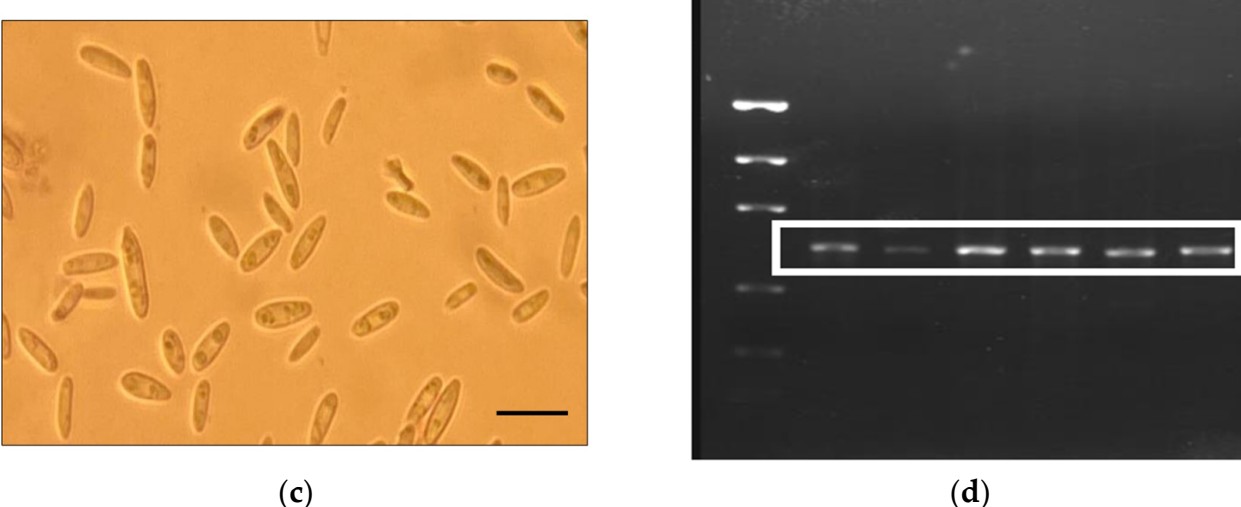

(c)                    (d)

**Figure 1.** Macroscopic and microscopic characteristics of *Fusarium musae* isolated from banana (*Musa × paradisiaca*). (**a**) Front colony on PDA plate; (**b**) back colony on PDA plate; (**c**) microconidia morphology (40×, Bar = 50 μm); (**d**) gel electrophoresis of PCR products amplified from DNA extracted from *Fusarium musae* (indicated in the white square).

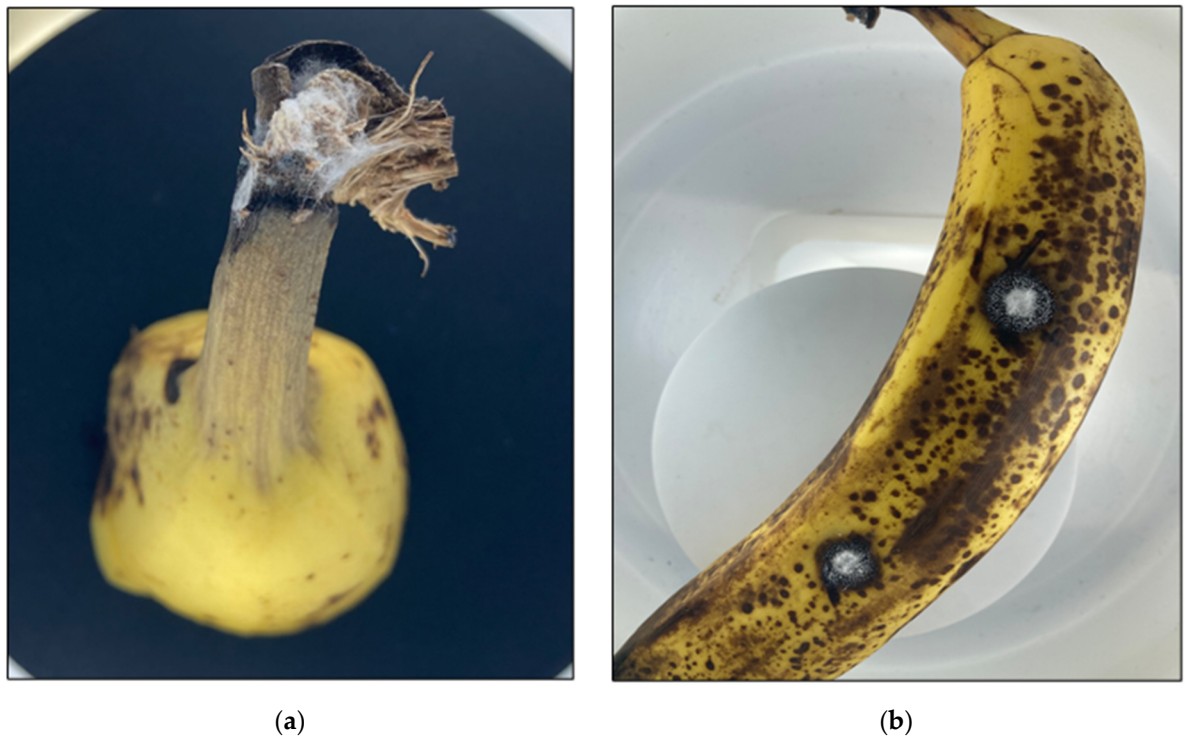

(a)                    (b)

**Figure 2.** Pathogenicity test results on banana fruit stored at ambient temperature with high relative humidity (90–95%) for nine days. (**a**) Crown rot; (**b**) fruit.

### 3.2. Chromatographic Analysis

The phenolic compounds found in the coconut mesocarp extract were gallic acid, protocatechuic acid, and chlorogenic acid (Figure 3), with retention times of 1.60 min, 2.44 min, and 3.86 min, respectively. There is little information on the profile of the phenolic compounds of coconut mesocarp by-products, which have reported the presence of epicatechin and 4-hydroxybenzoic acid, together with a smaller amount of ferulic acid [24].

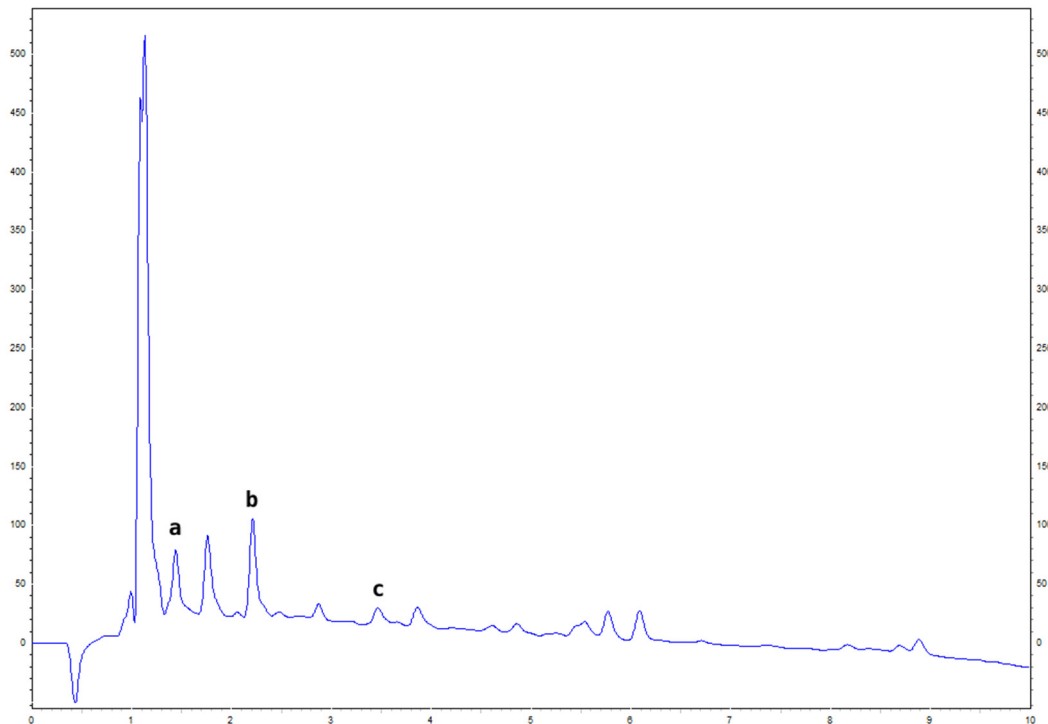

**Figure 3.** Chromatogram of the phenolic acids found in coconut mesocarp extract: (a) gallic acid; (b) protocatechuic acid; (c) chlorogenic acid.

### 3.3. In Vitro Antifungal Assay

*Fusarium musae* was successfully grown on the crown and fruit-based medium for six days at 25 °C (22% relative humidity). The mycelial development and spore germination were significantly different ($p < 0.05$) depending on the tissue (crown or fruit) and the extract concentration (Figure 4a,c). Overall, the extracts performed better on the crown-based medium than on the fruit medium using extracts at 5%. In vitro, assessments are crucial to determining the effectiveness of the plant extracts in a qualitative and quantitative way [19]. To our knowledge, this is the first time that in vitro conditions close to the conditions of *Fusarium musae* development on banana fruits have been simulated with a culture medium based on fruit tissues. Cortés-Rivera et al. [17] evaluated the inhibition of *P. italicum* in vitro in their investigation with the application of aqueous extracts from coconut mesocarp at 5%, and 76% mycelial inhibition was obtained. Conversely, in this investigation, only 26% was obtained using the same concentration of extracts. The germination process was strongly inhibited ($p < 0.05$), ranging from 81 to 91% compared to the control (0%) (Figure 4c). For the sporulation test, the treatments and the type of tissue (Figure 4b) were significant ($p < 0.05$). The results of this research suggest an important affectation of the mycelia, due to lower spore production ranging from $1.23 \times 10^4$ to $6.14 \times 10^4$ spores/mL compared to the control in the crown medium ($1.21 \times 10^5$ spores/mL) and fruit medium ($1.82 \times 10^5$ spores/mL) (Figure 4b).

### 3.4. In Vivo Antifungal Assay

The inhibition of the development of *Fusarium musae* on fruit is presented in Figures 5 and 6. At the end of storage, 100% of the control group developed the disease in crown and fruit tissues. Conversely, in treated bananas, the fungal decay was significantly minor, obtaining 58% (fruit) and 28% (crown) disease incidence (Figure 5a). The application of extracts was effective in reducing the infected wounds on the fruit by up to 40% (Figure 5b) and the severity of lesions by up to 40% (3 cm) (Figures 5c and 6d).

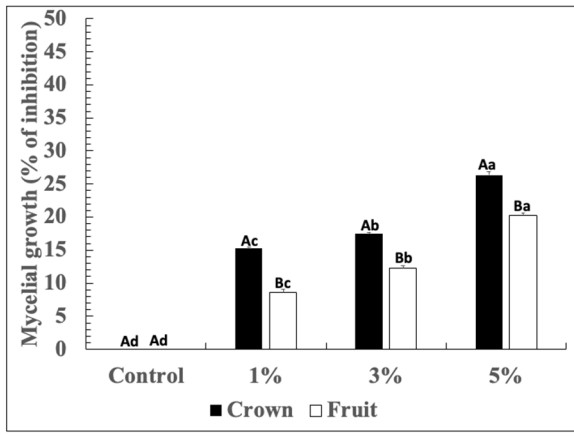

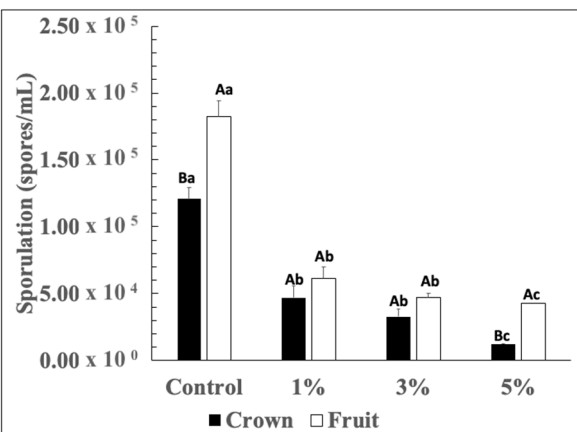

(**a**)  (**b**)

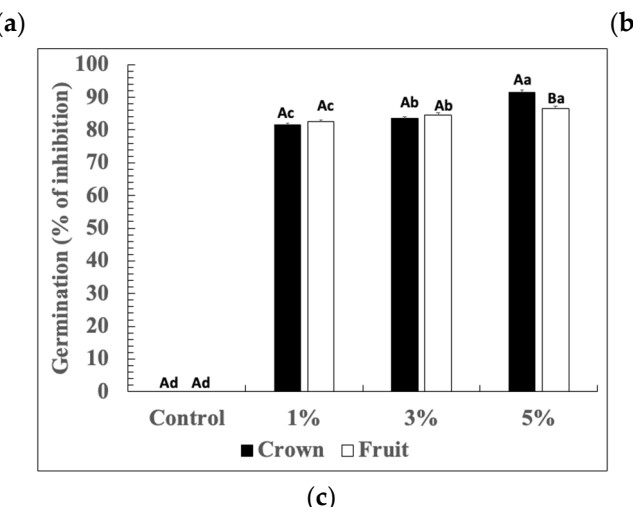

(**c**)

**Figure 4.** In vitro efficacy of treatments. (**a**) Mycelial development, (**b**) sporulation, and (**c**) germination against *Fusarium musae*. Different letters indicate significant differences between treatments (lowercase) and tissues (uppercase) according to the ANOVA and LSD test ($p < 0.05$).

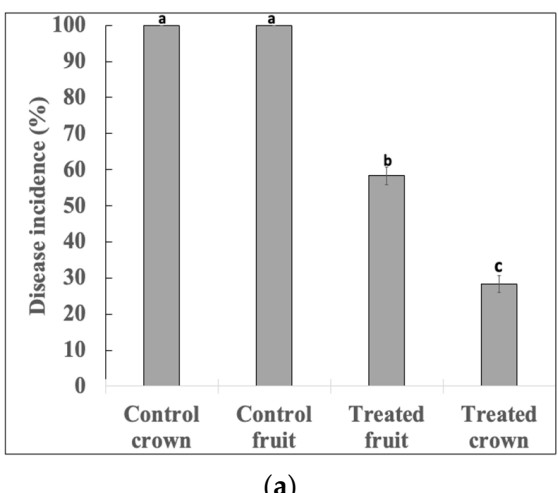

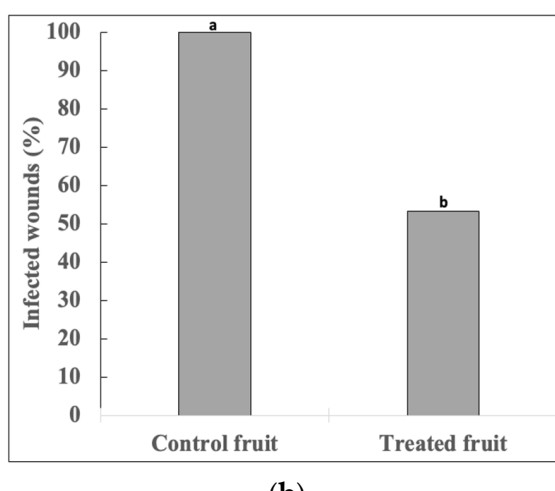

(**a**)  (**b**)

**Figure 5.** *Cont*.

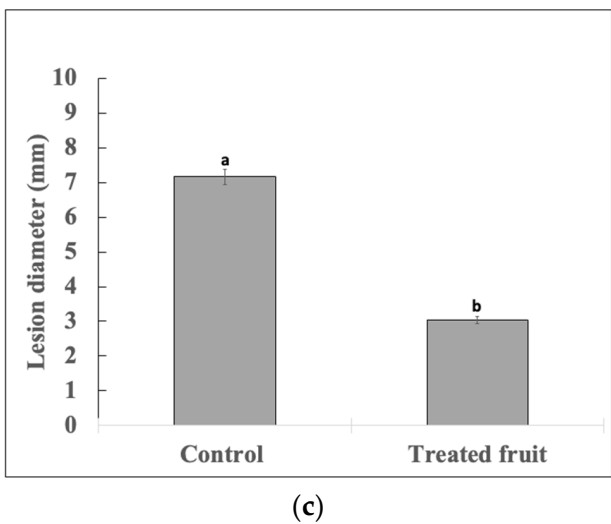

(**c**)

**Figure 5.** Efficacy of treatments applied on artificially infected banana fruits. (**a**) Disease incidence in crown and fruit, (**b**) percentage of infected wounds in fruit, and (**c**) severity in fruit. The fruit was treated with aqueous extract solutions at 5% and stored at 25 °C and 85–90% relative humidity for 6 days.

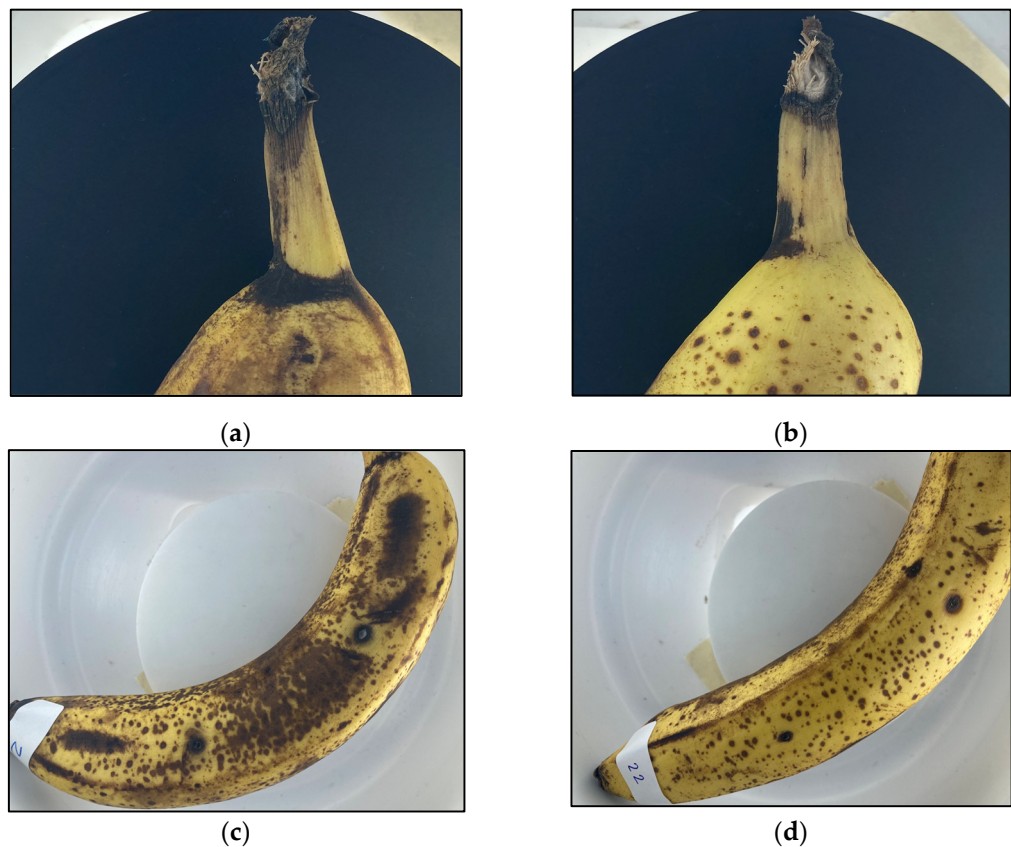

**Figure 6.** Inhibition of the development of *Fusarium musae*. (**a**) Crown without treatment (control, sprayed with SDW); (**b**) treated crown with aqueous extracts; (**c**) fruit without treatment (control, sprayed with SDW); (**d**) treated fruit with aqueous extracts. The fruit was treated with aqueous extract solutions at 5% and stored at 25 °C and 85–90% relative humidity for 6 days.

### 3.5. Scanning Electron Microscopy (SEM)

SEM observations showed extensive *Fusarium musae* development in the control group in the crown rot (Figure 7a) and fruit (Figure 7c) wounds. Conversely, non-germinated spores and a low presence of hyphae were observed on treated tissues (Figure 7b,d).

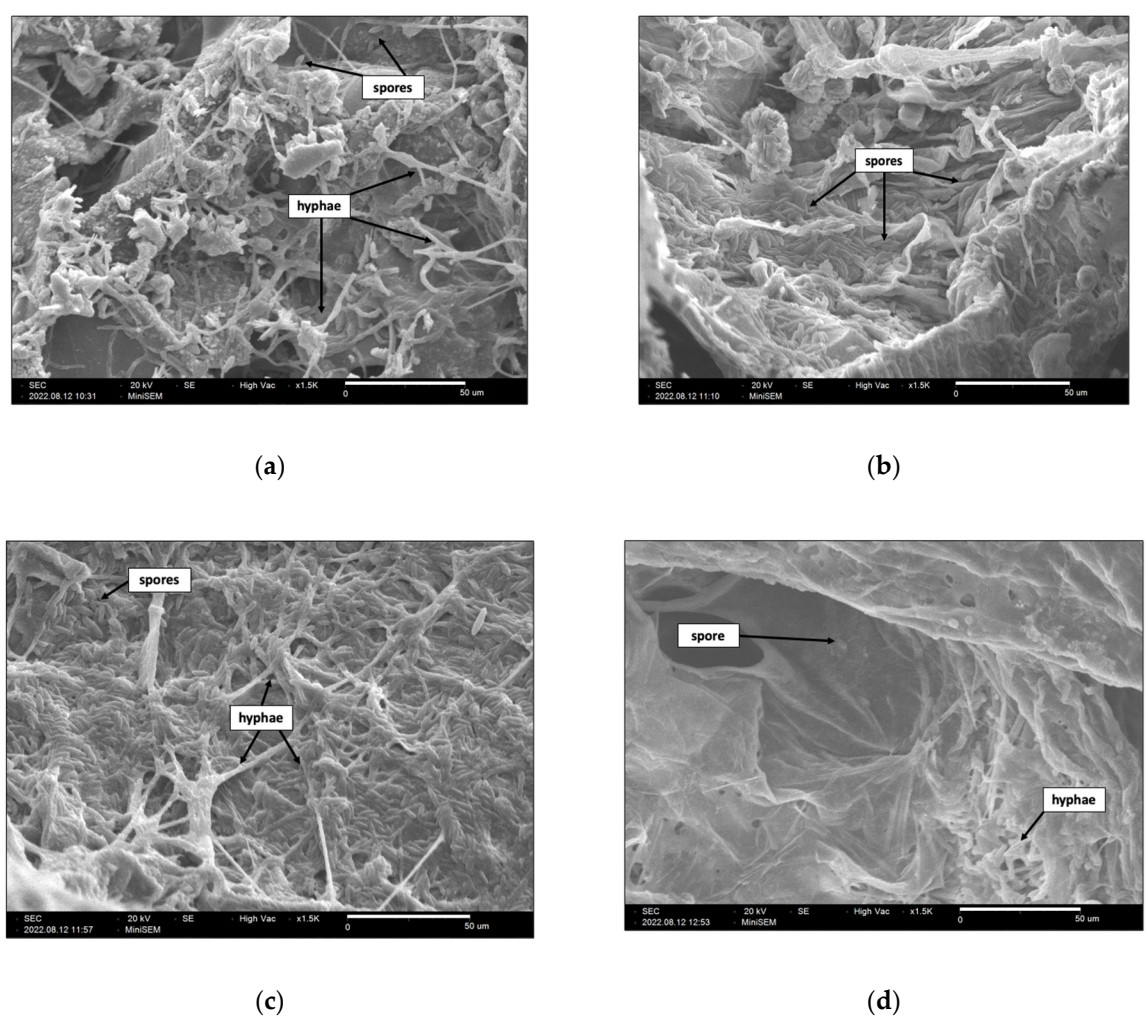

(**a**)  (**b**)

(**c**)  (**d**)

**Figure 7.** Scanning electron micrographs of treated fruits and fruits artificially infected with *Fusarium musae*. (**a**) Crown tissue without treatment (control, sprayed with SDW); (**b**) crown tissue treated with aqueous extracts; (**c**) fruit tissue without treatment (control, sprayed with SDW); (**d**) fruit tissue treated with aqueous extracts. Magnification of 1500×, bar = 50 μm.

### 4. Discussion

Cultural and morphological characteristics of *Fusarium musae* isolated in *Musa paradisiaca* similar to those in this study have recently been reported [25]. In the pathogenicity test conducted in this study, fruit rot was evident after three days of artificial infection with the presence of a necrotic area around the inoculation zone. These results are in agreement with Baria et al. [25]; in their study, they observed large areas of lesions up to the fourth day after inoculation.

In a recent study, compounds such as gallocatechin, coumaric, ferulic, and hydroxybenzoic acids, which were not identified in the present study, were identified in aqueous extracts of coconut mesocarp by-products [10], and the differences in the PC profile identified in the aqueous extracts may be due to different factors, including differences in the maturity and storage time of the coconut mesocarp by-products until their harvesting, and may even be due to the possible interactions that the PC could have between them, which could make it difficult to identify them in the HPLC-DAD equipment [26,27]; however, as

far as we know, the presence of gallic acid in coconut mesocarp by-products had not been reported, and this compound is strongly related to antifungal effects in vitro [28].

The compounds Identified in this study have been reported previously as antifungal agents [28–32]. Recently, Vázquez-González et al. [33] reported the mycelial inhibition of *Colletotrichum gloeosporioides* and *Penicillium italicum* by the use of a leaf extract from jackfruit in vitro tests, and in their study the effectiveness of plant extracts observed was associated to the phenolic compounds quinic acid, catechin, and chlorogenic acid. In general, it is well known that phenolic compounds can induce structural changes in membrane fungal cells affecting their fluidity, integrity, and permeabilization, and the alteration of these processes leads to the affectation of mycelial development [19,34].

The formation of germ tubes is crucial for fungal infections as a mechanism of penetration [35], and in the case of *Fusarium* spp. spores this process is crucial in the dissemination of the disease [36]. The results of the germination in this study are in agreement with Cortés-Rivera et al. [17], who reported that the inhibition of the germination of *P. italicum* was greater as the concentration of the extract increased. Furthermore, in another study, lower rates of germination were reported in *C. gloeosporioides* and *P. italicum* exposed to plant extracts rich in phenolic compounds such as chlorogenic acid [33]. The effectiveness of these compounds could be associated with negative changes in important biochemical processes and structural organization in the cell such as the disruption of the cell wall, alteration of membrane permeabilization (affecting its fluidity), depletion of adenosine triphosphate (ATP), and finally cell death [29,34].

The application of extracts at 5% was more effective by reducing up to 88% of the sporulation in the crown-based medium and the fruit-based medium (77%). The sporulation process of the genus *Fusarium* plays a key role in the disease cycle through spores, and in this sense, wounds caused by cutting off the hands after harvest are the point of access of spores to the crown; with favorable conditions, the spore can germinate, infect the fruit tissues, and induce necrosis and detachment of the fruit [36,37]. The in vitro results of this research are in agreement with the report by Cortés-Rivera et al. [17] against the blue mold agent in citrus and *Rhizopus stolonifer*, the soft rot agent in soursop [38]. The differences in the efficacy of treatments depending on the fruit tissue could be related not only to the nutrient composition of tissues [39] but also to the affectation of fungus due to the extract exposition in tissues, playing a key role in the fungus development as evidenced by SEM (Figure 7b,d).

The results of our in vivo assessments are in agreement with Hernández-Flores [10], who reported that coconut mesocarp extracts can control *P. italicum* under storage conditions of 25 and 13 °C, reducing the damage of treated fruit and the incidence of blue mold disease. These results open the spectrum of effectiveness of the coconut mesocarp extracts by inhibiting not only in vitro but also in vivo the important pathogens that attack a wide variety of fruit at the postharvest stage. As mentioned above, phenolic compounds can affect the normal development of phytopathogens by altering key functions of the membranes such as integrity, fluidity, and permeabilization [40]. In the case of gallic acid, its mechanism of action reported is associated with blocking the synthesis of chitin, an important component of the fungal cell wall, leading to its degradation [28]. Martínez et al. [29] reported that chlorogenic acid can induce fungal membrane permeabilization in *F. solani*, affecting not only the germination process and hyphae development but also the colonization of fruit tissues, as evidenced in this study by SEM (Figure 7b,d). The effect of hydroxycinnamic acid and its derivatives, such as protocatechuic acid, was tested against *F. oxysporum* f. sp. *niveum*, the causal agent of wilt disease in watermelon fruits, and the results showed a strong inhibition of mycelial development (63.7%) by using 1600 mg/L of the phenolic compound; furthermore, the sporulation process was suppressed up to 90%, and the germination decreased in a range from 40 to 100%. [41]. Nguyen et al. [32] reported that protocatechuic acid can affect the spore shape and germination process of a wide variety of pathogens such as *Botrytis cinerea*, *Fusarium oxysporum*, *Rihzoctonia solani*, and *Phytophthora capsici* during in vitro tests, and the authors evaluated the antifungal

potential of protocatechuic acid against *B. cinerea* in strawberry fruits. The results showed that protocatechuic acid can suppress gray mold on strawberry fruit 7 days after inoculation with *B. cinerea*, and the authors mentioned that the mechanisms involved can be associated with changes in membrane permeability and/or cell wall degradation, affecting important biochemical processes for fungi survival.

## 5. Conclusions

The use of aqueous extracts from coconut mesocarp was determined to be effective in controlling infection by *Fusarium musae*. The residues of coconut mesocarp are rich in phenolic compounds with important antifungal activity, and their use can impact the reduction of coconut residue accumulation in the environment. The use of coconut mesocarp extracts has been recently explored as an antifungal agent with promising results in limes and tomatoes. Thus, the information on the efficacy of this eco-friendly alternative for controlling pathogens in a variety of hosts is important due to the nature of the extraction, which uses only water and no solvents of chemical origin, helping to establish new and effective treatments for disease management.

**Author Contributions:** J.A.Y.M.-J.: writing—original draft, visualization, and investigation; F.J.B.-B.: investigation and validation; B.M.-L.: methodology and validation; L.G.H.-M.: formal analysis; P.U.B.-R: data curation; L.d.C.R.-I.: resources; P.G.-M.: supervision; R.R.G.-E.: validation, writing—original draft, writing—review and editing, supervision, project administration, and funding acquisition. All authors have read and agreed to the published version of the manuscript.

**Funding:** This research was funded by Tecnológico Nacional de México (grant number 9256.20-P).

**Data Availability Statement:** The datasets generated during the study are available from the corresponding authors upon reasonable request.

**Acknowledgments:** The authors are pleased to acknowledge Nancy D. Ruelas of UAN, who performed SEM analysis of the samples.

**Conflicts of Interest:** The authors declare no conflict of interest.

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
