# Peer review of "Coconut Mesocarp Extracts to Control Fusarium musae, the Causal Agent of Banana Fruit and Crown Rot"

_agriengineering, doi:10.3390/agriengineering5040147_

Round 1

Reviewer 1 Report

Comments and Suggestions for Authors

The article was carried out on a prioritised and important subject and with the joint work of different disciplines. For this reason, the study should be brought to the world of science from a scientific point of view and in terms of inspiration.

My suggestions are marked on the attached file (PDF) file.

However, there are some grammatical and scientific spelling mistakes in the article. These should be corrected. I think that these corrections should be written without time pressure. For this reason, I propose that the article be "rejected for rewriting and editing".

My suggestion; I think that the article will be read and cited more if it is re-prepared and published in the "MDPI Journal of Fungi (https://www.mdpi.com/journal/jof, ISSN: 2309-608X).

PCR image can be attached.

Colony grow, single spore or plant part

While the efficacy of "chlorogenic acid" is constantly mentioned, the efficacy of "gallic acid and Protocatechuic acid" is not referred to in the literature. Literature information on the efficacy of "gallic acid and Protocatechuic acid" should be given.

Comments on the Quality of English Language

 Minor editing of English language required

Author Response

Reviewer 1

The response to the revision is detailed as follows:

All changes done in the manuscript are marked in red color.

Line 3: the word “fruit” was added as suggested by the reviewer.

Lines 18-29: the corrections were made as suggested by the reviewer.

Line 31: the keywords were changed as suggested by the reviewer.

Lines 33-35: the sentence was corrected as suggested by the reviewer.

Lines 33-38: the corrections suggested were made.

Lines 40-42: the corrections suggested were made.

Lines 42-43: the information is about phytopathogens, not viruses or phytoplasma.

Lines 43-44: the corrections suggested were made.

Lines 45-50: the corrections suggested were made.

Reviewer comment: You need to write a method about the chromatographic analysis part.

Author´s response: the following information was added to the manuscript in lines 91-100 First, 0.1 g of the sample was dissolved in 5 mL of ultrapure water and vortexed for 1 h. Then, the sample was centrifuged (15 min at 5000 rpm) and filtered (0.45 μm membrane). Identification of phenolic compounds was performed in an HPLC-DAD equipment (Agilent 1200, Santa Clara, CA), using a Zorbax Eclipse Plus C18 reverse phase column (4.6 mm × 100 mm) with a particle size of 3.5 μm. The injection volume was 10.0 μL and the column temperature was 30 °C. A step gradient elution with absolute methanol (solvent A) and 1% formic acid (solvent B) was used: 18% A (0min); 30% A (4min); 45% A (8min); 55% A (10 min); 70% A (11 min) and 100% A (13 min). UV detection was carried out from 214 to 520 nm. The retention times of the standards were used for the identification of phenolic compounds.

Lines 54-55: the corrections suggested were made.

Line 67: the following reference was added to the manuscript Varela, G.B.; Jiménez, V.A.O.; Sañudo, R.B.; Martínez, P.G. Efecto Del Ácido Salicílico En La Inducción de Resistencia a Colletotrichum Sp. En Frutos de Plátano Durante Postcosecha. Revista Iberoamericana de Tecnología Postcosecha 2015, 16, 27–34.

Lines 60-80: the corrections suggested were made.

Line 73: the following reference was added to the manuscript Alvindia, D.G.; Natsuaki, K.T. Evaluation of Fungal Epiphytes Isolated from Banana Fruit Surfaces for Biocontrol of Banana Crown Rot Disease. Crop Protection 2008, 27, 1200–1207.

Line 107: the following reference was added in the manuscript, González Estrada, R.R.; Ascencio Valle, F. de J.; Ragazzo Sánchez, J.A.; Calderón Santoyo, M. Use of a Marine Yeast as a Biocontrol Agent of the Novel Pathogen Penicillium Citrinum on Persian Lime. Emir J Food Agric 2017, 29, 114–122, doi:10.9755/ejfa.2016-09-1273.

2.4 in vitro antifungal assay

Lines 103-133: the information was corrected as suggested by the reviewer.

Line 106: the citation previously mentioned in the paper, corresponds to the reference Ncama, K.; Mditshwa, A.; Tesfay, S.Z.; Mbili, N.C.; Magwaza, L.S. Topical Procedures Adopted in Testing and Application of Plant-Based Extracts as Bio-Fungicides in Controlling Postharvest Decay of Fresh Produce. Crop Protection 2019, 115, 142–151, doi:10.1016/j.cropro.2018.09.016. We decided the conditions (temperature and days) considering the necessary time for fungus development on PDA plates.

Line 108: in this part, we used only healthy tissues thus the crown is free of disease not rotten.

Line 110: the citation was corrected.

Line 128: this part, Subsequently, the treated fruit was dry (25 °C) for 1 h in a biosafety hood, was determined for us, taking into account the time needed for fruit drying in the the biosafety hood.

Line 138: we established that the SEM analysis would be performed at the end of the test.

Line 146: we use crowns disease free, they were artificially infected with Fusarium musae

Comment: In the discussion part; Discuss the colour and morphological structure with other Fusarium musae publications.In the Discussion section, discuss the 3 days with other Fusarium musae publications.

Response: The following paragraph was added to the manuscript “Similar cultural and morphological characteristics of Fusarium musae isolated in Musa paradisiaca to this study have recently been reported [25]. In the pathogenicity test conducted in this study, fruit rot was evident after three days of artificial infection, with the presence of a necrotic area around the inoculation zone, these results are in agreement with Baria et al. [25], in their study authors observed large areas of lesions to the 4th day after inoculation.” as suggested by the reviewer.

Comment: If a picture can be added, the subject will be proven. It is recommended to add a picture.

Response: The image of the PCR was added to the manuscript.

Comment: accession number OW986028, Citation is compulsory.

Response: The reference and citation with url were added to the manuscript.        NCBI (National Center for Biotechnology Information). Available online: https://www.ncbi.nlm.nih.gov/nuccore/OW986028 (3 November 2023).

Comment: Replace picture 1c with the new picture with scale.

Response: The picture 1c was changed as suggested by the reviewer.

Line 174: the pathogenicity test was carried out during 9 days, to obtain better images of the fungus development.

Comment: In "Figure 2" it is written that it is 9 days. It says 6 days here. Which is correct?

Response: the in vitro test was carried out for 6 days, as we commented above the pathogenicity test was carried out for 9 days to obtain better images of the fungus development.

Line 188-189: we used a healthy crown for the preparation of the culture medium, not rotten.

Line 203: the same comment above.

Comment: In Table 1 and Table 2 The addition of a control group (as a column) should be considered.

Response: the tables were changed to a figure to add the control and for a better representation of the data. We deleted the information about the % of spore reduction, instead, we mentioned in the text those values.

Line 220: we are only talking about the disease incidence, when we mention the severity then we link to the figs. 5 c,d.

Line 221: 50??? not 40,

response: the incidence of infected wounds expressed as a percentage was 53.33, thus the reduction is 46.57%

Line 222: the size of the severity of lesions was added to the manuscript.

Comment: Lines 268-269: The compounds identified in this study have been reported previously as antifungal agents [23–25]. Chlorogenic acid ok but 

Gallic acid, protocatechuic acid literature?

Response: we added the following references “Seo, D.-J.; Lee, H.-B.; Kim, I.-S.; Kim, K.-Y.; Park, R.-D.; Jung, W.-J. Antifungal Activity of Gallic Acid Purified from Terminalia Nigrovenulosa Bark against Fusarium Solani. Microb Pathog 2013, 56, 8–15.” and “Nguyen, X.H.; Naing, K.W.; Lee, Y.S.; Moon, J.H.; Lee, J.H.; Kim, K.Y. Isolation and Characteristics of Protocatechuic Acid from Paenibacillus Elgii HOA73 against Botrytis Cinerea on Strawberry Fruits. J Basic Microbiol 2015, 55, 625–634.”

Comment: Lines 282-286 While the efficacy of "chlorogenic acid" is constantly mentioned, the efficacy of "gallic acid and Protocatechuic acid" is not referred to in the literature. Literature information on the efficacy of "gallic acid and Protocatechuic acid" should be given.

Response: the following paragraph is in the manuscript “In the case of gallic acid, its mechanism of action reported is associated with blocking the synthesis of chitin, an important component of the fungal cell wall, leading to its degradation [28].”

“Nguyen et al. [32] reported that protocatechuic acid can affect the spore shape and ger-mination process of Botritys cinerea and F. oxysporum in vitro tests, besides authors evaluated the antifungal potential of protocatechuic acid against B. cinerea in strawberry fruits. The results showed that protocatechuic acid can suppress gray mold on strawberry fruit 7 days after inoculation with B. cinerea, authors mentioned that the mechanisms involved can be associated with changes in membrane permeability and/or cell wall degradation affecting important biochemical processes for fungi survival.”

Reviewer 2 Report

Comments and Suggestions for Authors

In this study the authors investigated the possibility of coconut mesocarp extract to control Fusarium musae. Overall, the work carried out is interesting and relevant to the present-day need. But it has not even done an acceptable amount of work. Following suggestions must be considered during the revision of the manuscript:

The general impression is that this study needs to be completed to have completeness. The material and the changes caused by the treatment should be examined by another method of characterization. in addition, a more detailed interpretation of the results is necessary.

2Choose more specific keywords related to this study.

 The introduction is too short. It is necessary to write a more extensive and detailed introduction. Furthermore, provide more detail about main aim of this study, highlight the novelty and scientific contribution of this study and state the methods used that led to the main findings of this study.

4Provide more information about banana collection (Part 2.1)

LList all chemicals used and their purity level (Part 2.1)

6 Indicate in the text whether the experiments were performed in multiple repetitions.

7Provide a description of the macro and micro analysis procedure (Section 2) .

8 Provide a description of statistical analysis (Section 2)

91.7 Statistical analysis correct in 2.7.

1A clearer and more detailed explanation of chromatographic analysis is needed.

1More detailed explanation of the SEM micrograph. Specifically describe the changes that occurred because of the treatment.

1Poor discussion of the results. Most of them are general statements. Supporting literature must be cited.

1 A more specific conclusion is needed that provides insight into the main results of this study and explains the scientific contribution of this study too.

Author Response

Reviewer 2

The response to the revision is detailed as follows:

All changes done in the manuscript are marked in red color.

Comment: The general impression is that this study needs to be completed to have completeness. The material and the changes caused by the treatment should be examined by another method of characterization. In addition, a more detailed interpretation of the results is necessary.

Response: The manuscript was improved as suggested by the reviewer, and the results and discussion sections were improved for a better understanding of the key results.

Comment: Choose more specific keywords related to this study.

Response: The key words were changed as suggested by the reviewer.

 Comment: The introduction is too short. It is necessary to write a more extensive and detailed introduction. Furthermore, provide more detail about main aim of this study, highlight the novelty and scientific contribution of this study and state the methods used that led to the main findings of this study.

Response: more studies were added to the introduction section about the use of coconut mesocarp extracts applied on limes and tomato fruits. The following paragraph was added to the manuscript “Recently, coconut mesocarp extracts were efficient in controlling Penicillium italicum in-fection in Persian lime fruits, by reducing the disease incidence and severity on artificially infected fruits [10]. Besides, in another study coconut mesocarp extracts as an additive were added into a chitosan matrix, and the results showed good biocompatibility of the extracts with the polymeric matrix by enhancing physicochemical and antifungal properties of films and coatings, protecting tomato fruits against Geotrichum candidum establishment [11].”

Comment: Provide more information about banana collection (Part 2.1)

Response: the following paragraph was moved to the part 2.1 “Banana fruits were purchased from a commercial orchard located at 21°28'28.6"N and 104°51'31.1"W in Tepic, Nayarit, Mexico.” as suggested by the reviewer.

Comment: List all chemicals used and their purity level (Part 2.1)

Response: the following paragraph was added to the text “Formic acid and methanol were purchased from Sigma Aldrich (St. Louis MO, USA).” to complete the list of chemicals used.

Comment: Indicate in the text whether the experiments were performed in multiple repetitions.

Response: the repetitions of the experiments are indicated in the 2.7 section, statistical analysis in lines 165-166.

Comment: Provide a description of the macro and micro analysis procedure (Section 2).

Response: the procedures were modified by adding more information and details, as suggested by the reviewer.

Comment:  Provide a description of statistical analysis (Section 2)

Response: the description was modified by adding more details, as suggested by the reviewer.

Comment: clearer and more detailed explanation of chromatographic analysis is needed.

Response: the procedure was modified by adding more information and details, as suggested by the reviewer.

Comment: A more detailed explanation of the SEM micrograph. Specifically describe the changes that occurred because of the treatment.

Response: the effects observed are detailed in section 3.5 and complemented with the discussion section, by connecting the results observed in the micrograph with the in vivo tests.

Comment: Poor discussion of the results. Most of them are general statements. Supporting literature must be cited.

Response: the discussion section was improved, including more results from the literature about the effect of phenolic compounds against fungi, as suggested by the reviewer.

Comment: A more specific conclusion is needed that provides insight into the main results of this study and explains the scientific contribution of this study too.

Response: the following paragraph was added to the manuscript “The use of aqueous extracts from coconut mesocarp was determined to be effective in controlling infection by Fusarium musae. The residues of coconut mesocarp are rich in phenolic compounds with important antifungal activity, their use can impact the reduction of coconut residue accumulation in the environment. The use of coconut mesocarp extracts has been recently explored as an antifungal agent with promising results in limes, and tomatoes. The use of coconut mesocarp extracts has been recently explored as an antifungal agent with promising results in limes, and tomatoes. Thus, the information on the efficacy of this eco-friendly alternative for controlling pathogens in a variety of hosts is important due to the nature of the extraction by using only water and no solvents from chemical origin, helping to establish new and effective treatments for disease management.” as suggested by the reviewer.

Reviewer 3 Report

Comments and Suggestions for Authors

1. In the introduction part, the authors should introduce the background about the coconut mesocarp extracts and plant disease control.

2. The "SEM analysis" and the "residue use " are not suitable as the keywords, they are experimental methods.

2. In the Figure 1, the bar for conidia picture should be added.

Author Response

Reviewer 3

The response to the revision is detailed as follows:

All changes done in the manuscript are marked in red color.

Comment: In the introduction part, the authors should introduce the background about the coconut mesocarp extracts and plant disease control.

Response: the following paragraph was added to the manuscript, “Recently, coconut mesocarp extracts were efficient in controlling Penicillium italicum in-fection in Persian lime fruits, by reducing the disease incidence and severity on artificially infected fruits [10]. Besides, in another study coconut mesocarp extracts as an additive were added into a chitosan matrix, and the results showed good biocompatibility of the extracts with the polymeric matrix by enhancing physicochemical and antifungal properties of films and coatings, protecting tomato fruits against Geotrichum candidum establishment [11].”

Comment: The "SEM analysis" and the "residue use " are not suitable as the keywords, they are experimental methods.

Response: the keywords were changed to “Keywords: SEM; Musa spp.; Crown rot; gallic acid; protocatechuic acid; chlorogenic acid”

Comment: In the Figure 1, the bar for conidia picture should be added.

Response: the bar was added as well as the scale.

Finally, the following references were added to the manuscript:

  1. Montes-Ramírez, P.; Montaño-Leyva, B.; Blancas-Benitez, F.J.; Bautista-Rosales, P.U.; Ruelas-Hernández, N.D.; Martínez-Robinson, K.; González-Estrada, R.R. Active films and coatings based on commercial chitosan with natural extracts addition from coconut by-products: physicochemical characterization and antifungal protection on tomato fruits. Food Control 2024, 155, 110077. https://doi.org/10.1016/j.foodcont.2023.110077
  2. Varela, G.B.; Jiménez, V.A.O.; Sañudo, R.B.; Martínez, P.G. Efecto del ácido salicílico en la inducción de resistencia a Colletotrichum sp. en frutos de plátano durante postcosecha. Revista Iberoamericana de Tecnología Postcosecha 2015, 16, 27–34.
  3. Alvindia, D.G.; Natsuaki, K.T. Evaluation of fungal epiphytes isolated from banana fruit surfaces for biocontrol of banana crown rot disease. Crop Protection 2008, 27, 1200–1207. https://doi.org/10.1016/j.cropro.2008.02.007
  4. Hernandez Montiel, L.G.; Zulueta Rodriguez, R.; Angulo, C.; Rueda Puente, E.O.; Quiñonez Aguilar, E.E.; Galicia, R. Marine yeasts and bacteria as biological control agents against anthracnose on mango. Journal of Phytopathology 2017, 165, 833–840. https://doi.org/10.1111/jph.12623
  5. Johnson, M.; Zaretskaya, I.; Raytselis, Y.; Merezhuk, Y.; McGinnis, S.; Madden, T.L. NCBI BLAST: A Better Web Interface. Nucleic Acids Res 2008, 36, W5–W9. 10.1093/nar/gkn201
  6. NCBI (National Center for Biotechnology Information). Available online: https://www.ncbi.nlm.nih.gov/nuccore/OW986028 (3 November 2023).
  7. Baria, T.T.; Rakholiya, K.B.; Chaudhari, A.K. Impact of inoculation methods and fruit maturity on development of fusarium fruit rot (Fusarium musae) disease in banana. Indian J. Agric. Res. 2021, 1, 4–7. doi:10.18805/IJARe.A-5731
  8. Nguyen, X.H.; Naing, K.W.; Lee, Y.S.; Moon, J.H.; Lee, J.H.; Kim, K.Y. Isolation and characteristics of protocatechuic acid from Paenibacillus elgii HOA73 against Botrytis cinerea on strawberry fruits. J Basic Microbiol 2015, 55, 625–634. DOI: 10.1002/jobm.201400041

Round 2

Reviewer 1 Report

Comments and Suggestions for Authors

The authors have made the suggested changes, but it is necessary for the sizes of the figures used in the manuscript to be the same. for example; it is not clear what the red arrow in the gel image is indicating. It is appropriate to present Figure 2a and 2b together as a single image rather than separately.

Comments on the Quality of English Language

 Minor editing of English language required.

Author Response

The size of the figures was modified, as suggested by the reviewer
The grammar of the manuscript was also checked and improved as suggested by the reviewer

Reviewer 2 Report

Comments and Suggestions for Authors

The authors corrected the manuscript acording to.reviewers suggestions

Author Response

Thanks for the comments and suggestions for the improvement of the manuscript